# SPR: A Structured Prompt Refinement Network for Modality Missing

Hao Chen [* 1 2]   Diwei Su [* 3]   Zhuo Wang [1]   Zuwang He [1]   Menglu Chen [§ 2]   Xiuxing Li [§ 1]   Xia Wu [1]

## Abstract

Prompt learning has recently emerged as a dominant paradigm to tackle the missing modalities challenge. However, existing methods often overlook the internal structural information of prompt vectors, limiting performance in guiding frozen backbone models under diverse missing modality scenarios. To address this, we propose a **S**tructured **P**rompt **R**efining (**SPR**) network that refines the internal structure of prompt vectors across multiple dimensions: (1) a Global Interaction Fusion Module captures bidirectional interactions across prompt layers, thereby mitigating suboptimal adaptation from inconsistent guidance under missing modalities, (2) a Local Feature Refinement Module structures adjacent prompt vectors into coherent semantic units, leveraging local contextual relationships to maintain semantic integrity during modality absence, and (3) a Channel Feature Selection Module uses point-wise gating to adaptively suppress noise and enhance critical channels based on the specific missing modality. Using only 0.8% trainable parameters, SPR achieves significant improvements on three mainstream multimodal classification datasets. Notably, it surpasses state-of-the-art by 3.8% in F1-Macro on the MM-IMDB dataset, even at a 90% modality missing rate. Extensive experiments and in-depth ablations validate SPR's effectiveness and robustness under various missing conditions.

## 1. Introduction

Multimodal learning, which integrates and processes heterogeneous data sources such as vision, language, and audio,

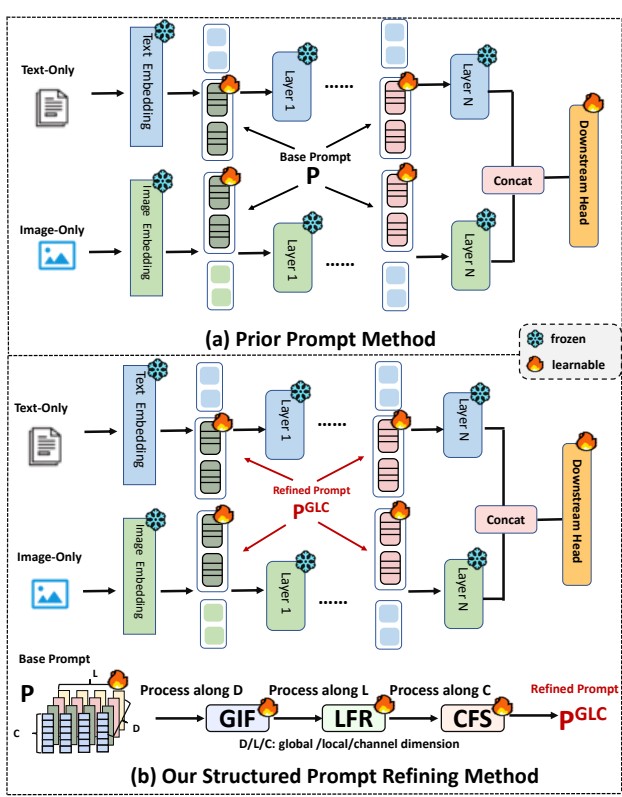

*Figure 1.* Prior prompt-based methods vs. our SPR in tackling incomplete multimodal learning. Prior prompt methods ignore prompt structural features, while our SPR explicitly refines them along three dimensions.

has become a cornerstone of modern artificial intelligence (Cheng et al., 2025; Liu et al., 2025), achieving state-of-the-art performance in numerous applications including path planning (Jia et al., 2025; 2024; 2026), sentiment analysis (Wang et al., 2025; 2024), and medical diagnostics (Wang et al., 2020; Ramazanova et al., 2025; Lu et al., 2025; Liao et al., 2025). However, its practical deployment is often hindered by missing modalities—data unavailable due to sensor failure (Ma et al., 2022; Wang et al., 2023), privacy constraints, or acquisition costs. This absence disrupts inter-modal correlations and degrades performance. To address the challenge of incomplete multimodal data, various research paradigms have emerged. Early approaches focused primarily on either data reconstruction or shared representa-

[*]Equal contribution [§]Corresponding author [1]School of Computer Science and Technology, Beijing Institute of Technology, Beijing, China [2]School of Optoelectronics, Beijing Institute of Technology, Beijing, China [3]School of Artificial Intelligence, Beijing Normal University, Beijing, China. Correspondence to: Xiuxing Li <xxl@bit.edu>, Menglu Chen <menglu@bit.edu.cn>.

*Proceedings of the 43rd International Conference on Machine Learning, Seoul, South Korea. PMLR 306, 2026. Copyright 2026 by the author(s).*

tion learning. Reconstruction methods (Wu et al., 2024; Ma et al., 2021; Luo et al., 2023; Dai et al., 2025) attempt to synthesize the missing modalities before feeding the data to a downstream classifier. However, such methods are often computationally expensive and risk introducing misleading artifacts that can impair performance. Alternatively, shared representation learning methods aim to learn a shared latent space, projecting both complete and incomplete data into a common representation that is robust to missingness (Cui et al., 2025; Geraghty et al., 2025; Qian et al., 2025; Lee & Pavlovic, 2020; Du et al., 2018). Yet, forcing diverse modal combinations into a single space often leads to the loss of crucial modality-specific details, thereby constraining the model's expressive power (Lee & Pavlovic, 2021; Zhao et al., 2024).

Prompt-based learning has emerged as an efficient paradigm, achieving state-of-the-art results by freezing the backbone network and fine-tuning only small prompts for incomplete data (Yang et al., 2022b; Khattak et al., 2023). Current methods primarily employ shallow networks or dynamic adapters to generate prompts based on unidirectional layer-wise correlations (Zhang et al., 2025b; Shi et al., 2024). These approaches neglect the intrinsic structural dependencies within the prompt sequence, particularly the global cross-layer correlations and local contextual consistency among prompt tokens (see Fig. 1). In contrast, our proposed SPR network explicitly refines these internal structures across global, local, and channel dimensions, thereby generating semantically coherent and adaptive prompts that robustly guide the frozen backbone.

To overcome these limitations, we propose the SPR network, which synergistically refines the internal structure of prompt vectors across three complementary dimensions. First, to address the lack of cross-layer collaboration, the **Global Interaction Fusion (GIF)** module utilizes a Bidirectional Recurrent Network with residual connections. This captures bidirectional associations between prompt layers, balancing high-level semantics with low-level features to resolve the suboptimal adaptation problem that arises when traditional prompt learning methods address missing modality tasks. Complementing this global perspective, the **Local Feature Refinement (LFR)** module employs depthwise convolutions to model local contextual relationships within each layer, transforming independent tokens into coherent semantic units to maintain consistency during modality absence. Finally, to achieve fine-grained adaptation, the **Channel Feature Selection (CFS)** module applies a pointwise gating mechanism along the channel dimension. This dynamically amplifies or suppresses specific feature dimensions, such as distinctively filtering text-related channels when the text modality is missing, thereby endowing the prompts with rich structural information to robustly guide the frozen backbone. Processing through these three modules endows the prompt

vectors with rich structural information, significantly enhancing their capability to guide the frozen backbone model, particularly under diverse modality missing scenarios. Our main contributions are as follows:

- We propose the SPR network, the first PEFT framework to address the missing modality problem by explicitly modeling and refining the prompt's internal structure. This addresses the limitation of prior state-of-the-art methods where prompt vectors lacked internal structural information.

- We design three innovative and lightweight modules, including Global Interaction Fusion, Local Feature Refinement, and Channel Feature Selection, that work in synergy to create highly adaptive and robust prompt representations.

- Extensive experiments show that SPR sets a new state-of-the-art on three benchmark datasets. Notably, on the MM-IMDb (Arevalo et al., 2017) dataset with a 90% missing rate, SPR outperforms the previous best method by 3.8% on the F1-Macro score, using only 0.8% of the total parameters.

## 2. Related Work

### 2.1. Approaches to Missing Modality Learning

Addressing the challenge of missing modalities, research has diverged into two main paradigms. One is Generative Reconstruction, which has evolved from early GANs (Salimans et al., 2016) and VAEs (Tomczak & Welling, 2018) to state-of-the-art diffusion models (Zhang et al., 2025a) for high-fidelity imputation. However, this paradigm is computationally expensive, prone to introducing misleading artifacts (Zhang et al., 2022; Ma et al., 2021; Sharma & Hamarneh, 2019), and has been shown to suffer from modality generation bias (Dai et al., 2025). The other paradigm, Shared Latent Space Learning, avoids reconstruction by aligning complete and incomplete data in a common space (Kim et al., 2025; Yang et al., 2022a; Yao et al., 2024). Yet, this forced alignment imposes an implicit constraint on intra-class representation, limiting the model's ability to capture modality-specific details (Lee & Pavlovic, 2021; Li et al., 2024). Given the inherent limitations of these approaches, prompt-based learning has emerged as an efficient alternative.

### 2.2. Prompt Learning for Missing Modality

Prompt-based learning has emerged as the leading parameter-efficient fine-tuning strategy for adapting large, frozen backbone models to missing modality scenarios (Zhang et al., 2025b; Yang et al., 2022b; Zhang et al., 2024). The evolution of this sub-field has been rapid, moving from

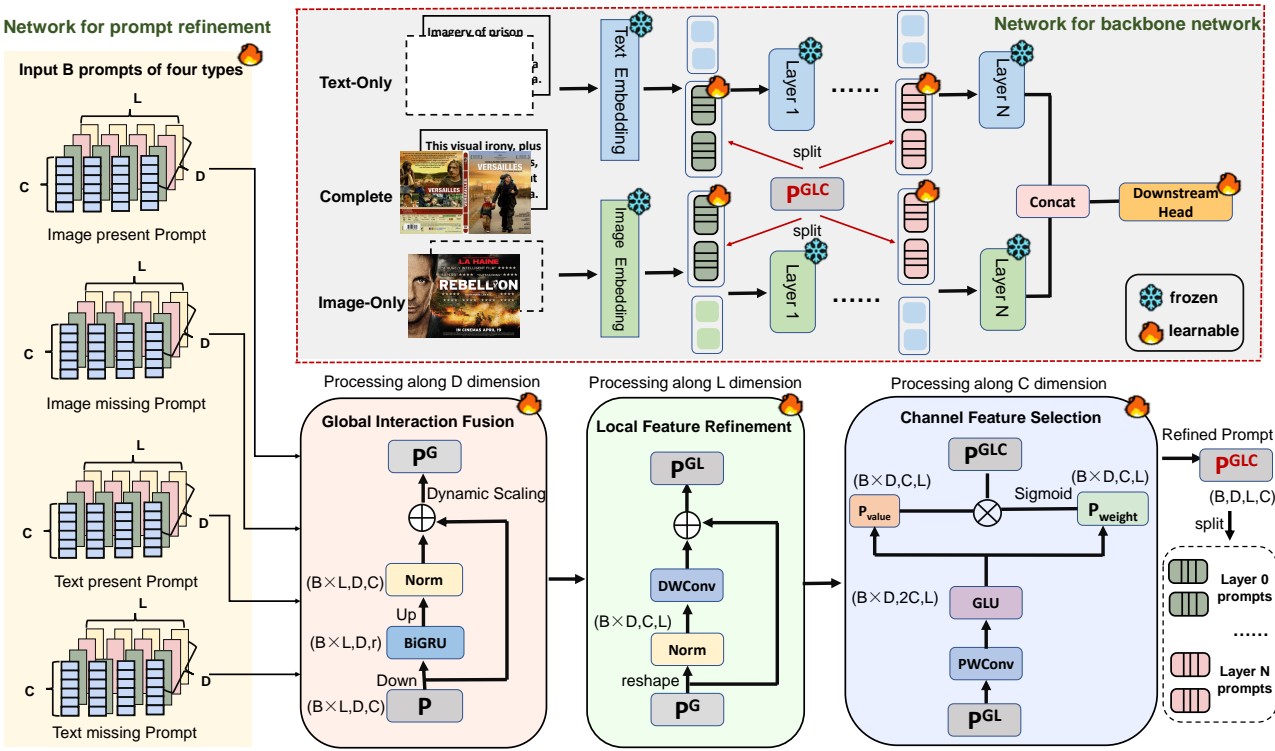

Figure 2. **Overall Framework of Our proposed framework.** The framework comprises two distinct components: the SPR network for prompt refinement and the frozen pre-trained backbone (CLIP (Radford et al., 2021)). For diverse missing scenarios, four types of initial prompts are constructed based on modality availability. These prompts undergo hierarchical refinement across Depth ( $D$ ), Length ( $L$ ), and Channel ( $C$ ) dimensions, generated via Global Interaction Fusion, Local Feature Refinement, and Channel Feature Selection modules, respectively. The final structured prompt $\mathbf{P^{GLC}}$ is injected into the backbone layers to interact with frozen image and text embeddings. The output features are concatenated and passed through a downstream head for prediction. During training, only the SPR network and the downstream head are updated, while the pre-trained backbone remains frozen.

static to dynamic prompt generation. The foundational work in this area, MMP (Lee et al., 2023), introduced missing aware prompts, a static set of learnable vectors that guide the model. A critical limitation of this method is that it employs static prompts that remain constant across different input samples. To overcome this, recent state-of-the-art methods have shifted entirely to dynamic prompts generated on-the-fly for each input. These methods fall into two main categories. The first category is retrieval-augmented generation, exemplified by RAGPT (Lang et al., 2025), which utilizes a context-sensitive prompter driven by a retrieval mechanism, and MemPrompt (Zhao et al., 2025), which extracts generative prompts from a fixed memory bank. The second category computes prompts directly from input features. This includes SyP (Zhang et al., 2025b), which uses a dynamic adapter to compute scaling factors, and MGR (Huang et al., 2025), which generates mixed-context prompts. Despite this progress, a crucial and shared limitation persists. Previous state-of-the-art methods, such as SyP (Zhang et al., 2025b), RAGPT (Lang et al., 2025), and even the most structure-focused DCP (Shi et al., 2024), mainly establishes a one-way correlation between prompt vectors at differ-

ent layers by using a shallow MLP to generate the current layer's prompt based on the preceding one. To overcome this limitation, we propose the Structured Prompt Refining network to perform explicit prompt refinement across three structural dimensions.

## 3. Methodology

### 3.1. Overall framework

**Problem Definition**. We formulate the task of learning with incomplete multimodal data. Without loss of generalizability, we primarily consider a scenario with $M = 2$ modalities, $m_1$ (e.g., text) and $m_2$ (e.g., image). Let $D = \{D_c, D_{m_1}, D_{m_2}\}$ be the full dataset, which comprises three distinct subsets based on modality availability. We denote $D_c = \{(x_{m_1}, x_{m_2}), y\}$ as the modality-complete subset, where $x_{m_1}$ and $x_{m_2}$ are the feature representations for $m_1$ and $m_2$ , and $y$ is the label. The missing-modality subsets are $D_{m_1} = \{(x_{m_1}), y\}$ , where $x_{m_2}$ is absent, and $D_{m_2} = \{(x_{m_2}), y\}$ , where $x_{m_1}$ is absent. Our goal is to develop a single, robust learning framework $f$ that can

effectively process inputs from any subset— $(x_{m_1}, x_{m_2})$, $(x_{m_1})$, or $(x_{m_2})$ —to produce accurate predictions.

Fig. 2 illustrates the core modules of SPR and their interconnections. We adopt a CLIP (Radford et al., 2021) as the backbone to leverage rich semantic priors. Input text and image are first processed by frozen embedding layers into token sequences. To address missing modalities, we initialize four types of prompts (e.g., image-missing prompt or text-present prompt) based on the available modality. These prompts are refined by the SPR network across Depth, Length, and Channel dimensions. The resulting refined prompts $\mathbf{P}^{GLC}$ are injected into the backbone layers to guide the frozen features. Finally, the outputs are concatenated and passed through a downstream head for classification. During training, only the SPR network and the head are updated, while the backbone remains frozen. The following sections delve into the specifics of each component and their respective implementations.

## 3.2. Global Interaction Fusion Module

MMP (Lee et al., 2023) employs independent prompts, inserted at the input and intermediate stages of the multimodal backbone, to direct model predictions. While this can theoretically provide sufficient guidance for features in each layer, the prompts across layers lack synergy. Subsequently, DCP (Shi et al., 2024) utilizes a simple MLP network to generate the current layer's prompt based on the prompt from the preceding layer. However, this relationship is unidirectional. To address this limitation, we propose the Global Interaction Fusion module.

This module utilizes a bidirectional recurrent network to capture dependencies axially along the layer dimension. Let the basic prompts for the image and text modalities be denoted as $\mathbf{P} \in \mathbb{R}^{B \times D \times L \times C_{\mathrm{m}}}$. Here, $B$ is the batch size, $D$ is the number of layers, $L$ is the number of prompt tokens per layer, and $C$ is the modality-specific embedding dimension. To decrease the required parameters and add non-linearity, we adopt a bottleneck design. Taking the image modality as an example, the reshaped prompts $\mathbf{P}' \in \mathbb{R}^{(B \times L) \times D \times C_{\mathrm{m}}}$ are first projected into a shared low-rank space $r$:

$$\mathbf{Z}_{\mathrm{m}} = \mathrm{ProjDown}_{\mathrm{m}}(\mathbf{P}') \qquad (1)$$

where $\mathrm{ProjDown}_{\mathrm{m}}(\cdot)$ is a modality-specific linear layer mapping $C_{\mathrm{m}} \to r$. Then we apply a shared, single-layer Bidirectional GRU (BiGRU) along the $D$ dimension to capture hierarchical dependencies:

$$\mathbf{H} = \mathcal{F}_{\mathrm{BiGRU}}(\mathbf{Z}_{\mathrm{m}}) \qquad (2)$$

The $\mathcal{F}_{\mathrm{BiGRU}}(\cdot)$ module has an input size of $r$ and a hidden size of $r/2$, and its bidirectional nature results in an output $\mathbf{H} \in \mathbb{R}^{(B \times L) \times D \times r}$. This recurrent core is shared by both image and text modalities to learn a generalized

layer correlation mapping. The context-aware features $\mathbf{H}$ are then projected back to the original dimension using a corresponding up-projection layer:

$$\mathbf{Y}_{\mathrm{m}} = \mathrm{ProjUp}_{\mathrm{m}}(\mathbf{H}) \qquad (3)$$

where $\mathbf{Y}_{\mathrm{m}} \in \mathbb{R}^{(B \times L) \times D \times C_{\mathrm{m}}}$. This output is then reshaped to its original $(B, D, L, C_{\mathrm{m}})$ format. Finally, the module integrates a simple and efficient gated residual connection to allow the model to adaptively balance its reliance on the original Basic Prompts $\mathbf{P}$ and the new Contextual Prompts $\mathbf{Y}$. The final refined prompt, $\mathbf{P}^{\mathrm{G}}$, is computed as:

$$\mathbf{P}^{\mathrm{G}} = \mathrm{LayerNorm}(\mathbf{P} + \alpha \cdot \mathbf{Y}) \qquad (4)$$

Here, $\alpha$ is a single, learnable scalar parameter, obtained via $\alpha = \sigma(\alpha_{\mathrm{logit}})$, which is broadcast across all dimensions. This scalar gate adaptively scales the contribution of the hierarchical context $\mathbf{Y}$. This approach mitigates the suboptimal adaptation problem stemming from the lack of collaboration among cross-layer prompts.

## 3.3. Local Feature Refinement Module

Following the Global Interaction Fusion Module, the refined prompts $\mathbf{P}^{\mathrm{G}}$ possess strong inter-layer relationships. However, the vectors within each layer are still treated as independent tokens. This overlooks the local contextual relationships between adjacent prompt vectors, limiting their collective ability to form a coherent semantic instruction. We argue that prompts should possess local structure, enabling them to evolve from a set of independent tokens into cohesive semantic units. Thus, we propose a Local Feature Refinement Module to explicitly model these local relationships.

This module is implemented using a lightweight, shared depthwise convolution with a kernel size of c. This operation is applied axially along the $L$ dimension (the sequence of prompt tokens) of the hierarchically refined prompts $\mathbf{P}^{\mathrm{G}}$. Specifically, the depthwise convolution $\mathcal{F}_{\mathrm{DWConv}}(\cdot)$ processes each channel $c \in [1, \ldots, C]$ independently, forcing each prompt vector to aggregate information from its immediate neighbors within the same layer. To maintain parameter efficiency and learn a generalized structural representation, this $\mathcal{F}_{\mathrm{DWConv}}(\cdot)$ module is shared across both the image and text modalities. The final prompt representation $\mathbf{P}^{\mathrm{GL}}$ is computed as:

$$\mathbf{P}^{\mathrm{GL}} = \mathrm{LayerNorm}(\mathbf{P}^{\mathrm{G}} + \mathcal{F}_{\mathrm{DWConv}}(\mathbf{P}^{\mathrm{G}})) \qquad (5)$$

We integrate this operation within a residual connection, followed by a LayerNorm, to ensure stable training. This transformation allows the prompts injected into the backbone to evolve from independent tokens into semantic units with meaningful local structures. Such a transition ensures

that a high degree of semantic consistency is maintained, which is critical for robustness in modality-missing scenarios. This enhancement of the local structure significantly improves the representational capacity of the prompts.

### 3.4. Channel Feature Selection

Following the Local Feature Refinement Module, the prompts $\mathbf{P}^{\text{GL}}$ possess robust hierarchical and local structures. However, the feature representation *within* each individual prompt vector is still treated as a monolithic block. The model lacks a mechanism to dynamically re-weight these internal features based on the context, such as a missing modality. We argue that for maximal adaptivity, the prompt system must perform fine-grained feature selection within each vector. This allows the prompt to specialize its representation based on the specific modality-missing condition. Thus, we propose a Channel Feature Selection Module.

This module applies a fine-grained, data-driven gating mechanism to the channel dimension of each prompt vector. The core of this module is a shared, pointwise (1x1) convolution followed by a Gated Linear Unit (GLU) (Shazeer, 2020). We first reshape $\mathbf{P}^{\text{GL}}$ to a tensor of shape $(B \times D, C, L)$ to apply the 1D convolution efficiently across the $L$ tokens. Let this be $\mathbf{P}'$. A shared Conv1d with kernel_size=1 is applied, which projects the channel dimension $C$ to $2C$. This operation is shared for both image and text modalities:

$$\mathbf{Y} = \mathcal{F}_{\text{PWConv}}(\mathbf{P}') \tag{6}$$

where $\mathbf{Y} \in \mathbb{R}^{(B \times D) \times 2C \times L}$. The tensor is then transposed to $\mathbb{R}^{(B \times D) \times L \times 2C}$ to isolate the channel dimension for the gating operation. Next, we apply the GLU. The tensor $\mathbf{Y}$ is split into two equal halves along the channel dimension, $\mathbf{P}_{\text{value}}$ and $\mathbf{P}_{\text{weight}}$, both of shape $\mathbb{R}^{(B \times D) \times C \times L}$. The final gated output is computed as:

$$\mathbf{P}^{\text{GLC}} = \mathbf{P}_{\text{value}} \odot \sigma(\mathbf{P}_{\text{weight}}) \tag{7}$$

Here, $\odot$ denotes element-wise multiplication and $\sigma(\cdot)$ is the sigmoid function. This GLU mechanism allows the module to learn a dynamic, non-linear selection gate $\sigma(\mathbf{P}_{\text{weight}})$ for the feature-space $P_{\text{value}}$. This process enables the model to selectively amplify useful features and suppress irrelevant ones based on the specific modality condition.

## 4. Experiment

### 4.1. Experimental Settings

**Datasets**. Following the experimental protocols established in MMP (Lee et al., 2023), DCP (Shi et al., 2024), and SyP (Zhang et al., 2025b), we conduct evaluations on three diverse benchmarks to assess generalization across different multimodal challenges. **MM-IMDb** (Arevalo et al., 2017)

comprises 25,959 movie poster-plot pairs annotated with 23 genres, serving as a testbed for multi-label classification where the semantic informativeness of each modality varies significantly across categories. To evaluate robustness against real-world noise, we employ **UPMC Food-101** (Wang et al., 2015), which contains 90,840 samples spanning 101 fine-grained categories characterized by uncurated web descriptions. The **Hateful Memes** dataset (Kiela et al., 2020) targets binary hate speech detection by utilizing benign confounders that invalidate unimodal priors, thereby necessitating deep cross-modal reasoning to capture subtle semantic conflicts between visual and textual content.

**Implementation details**. We utilize the CLIP (Radford et al., 2021) as our multimodal backbone, leveraging the ViT-B/16 (Hong et al., 2024) model as the image encoder. In line with the standard CLIP (Radford et al., 2021), all input images are uniformly resized to a 224x224 resolution. For the textual modality, we employ the tokenizer from the pretrained CLIP (Radford et al., 2021) model to process text inputs, which are subsequently truncated or padded to a maximum sequence length of 77 tokens. Our training adopts a parameter-efficient strategy. Specifically, we freeze the parameters of both the image and text encoders. Optimization is confined solely to the parameters of our deep correlated prompts and the final task-specific fully-connected (fc) layer. We configure the length of these learnable prompts, $L_p$, to 40. These prompts are prepended to the input features at $M = 6$ distinct layers within the transformer architecture. For optimization, we use the AdamW (Zhuang et al., 2022) with an initial learning rate of 1e-2 and a weight decay of 2e-2. The learning rate schedule includes a warmup phase for the initial $10\%$ of total training steps, followed by a linear decay to zero. All experiments were conducted with a batch size of 36 on a single NVIDIA L40S GPU.

**Metrics**. Given the distinct nature of the classification challenges across these datasets, we employ specific and appropriate evaluation metrics for each one. For **MM-IMDb** (Arevalo et al., 2017), F1-Macro is adopted to measure the multi-label classification performance; For **UPMC Food-101** (Wang et al., 2015), the metric is the classification accuracy; For **Hateful Memes** (Kiela et al., 2020), we use AUROC.

**Setting of Missing Modality**. Following MMP (Lee et al., 2023), DCP (Shi et al., 2024), SyP(Zhang et al., 2025b), we quantify missing modalities via a missing rate $\eta$. We define three bimodal settings: **Missing-Both** ($\frac{\eta}{2}$ text-only, $\frac{\eta}{2}$ image-only), **Missing-Text** ($\eta$ image-only), and **Missing-Image** ($\eta$ text-only), with the remaining $(1 - \eta)$ being complete. This protocol extends to $M$ modalities by assigning $\frac{\eta}{M^2-2}$ of the data to each incomplete configuration. Unless stated otherwise, $\eta = 70\%$.

*Table 1.* Comparison with CoOp (Zhou et al., 2022), MMP (Lee et al., 2023), MaPLe (Khattak et al., 2023), DePT (Zhang et al., 2024), DCP (Shi et al., 2024) and SyP (Zhang et al., 2025b) on the MM-IMDb, UPMC Food-101, and Hateful Memes datasets under various missing-modality cases with different missing rates. The bold number indicates the best performance.

| Datasets | | Train/Test | | Validation set | | | | | | | Testing set | | | | | | |
|---|---|---|---|---|---|---|---|---|---|---|---|---|---|---|---|---|---|
| | | Image | Text | CoOp | MMP | MaPLe | DePT | DCP | SyP | **Ours** | CoOp | MMP | MaPLe | DePT | DCP | SyP | **Ours** |
| MM-IMDb (F1-Macro) | 50% | 100% | 50% | 51.23 | 52.07 | 52.76 | 53.87 | 55.23 | 56.16 | **57.35** | 48.06 | 48.88 | 49.58 | 50.64 | 52.13 | 53.90 | **55.73** |
| | | 50% | 100% | 53.04 | 54.52 | 55.26 | 56.04 | 57.32 | 58.55 | **59.21** | 49.89 | 51.46 | 52.32 | 52.78 | 54.32 | 56.27 | **57.52** |
| | | 75% | 75% | 51.46 | 52.12 | 52.87 | 54.02 | 55.45 | 56.81 | **58.12** | 48.37 | 49.32 | 49.56 | 50.87 | 52.32 | 55.02 | **56.92** |
| | 70% | 100% | 30% | 47.26 | 48.23 | 48.75 | 49.87 | 51.35 | 53.34 | **54.11** | 44.13 | 45.64 | 45.52 | 46.38 | 48.52 | 51.37 | **53.58** |
| | | 30% | 100% | 52.32 | 53.21 | 53.98 | 55.04 | 56.21 | 56.89 | **57.21** | 48.82 | 50.52 | 50.64 | 52.13 | 53.14 | 54.20 | **55.91** |
| | | 65% | 65% | 50.22 | 51.34 | 52.31 | 53.17 | 54.24 | 55.22 | **56.37** | 46.84 | 48.12 | 49.16 | 50.32 | 51.42 | 52.90 | **54.95** |
| | 90% | 100% | 10% | 47.86 | 48.84 | 50.12 | 50.98 | 52.36 | 53.47 | **54.52** | 44.76 | 45.32 | 46.84 | 47.56 | 49.26 | 50.21 | **51.37** |
| | | 10% | 100% | 51.65 | 52.36 | 53.14 | 54.12 | 55.42 | 56.69 | **57.81** | 48.32 | 49.12 | 50.13 | 50.88 | 52.22 | 53.72 | **55.19** |
| | | 55% | 55% | 47.44 | 48.04 | 48.82 | 49.98 | 51.26 | 53.06 | **54.61** | 44.12 | 44.87 | 45.12 | 46.54 | 48.04 | 49.63 | **53.41** |
| Food101 (Accuracy) | 50% | 100% | 50% | 77.36 | 78.24 | 79.87 | 80.24 | 82.33 | 83.47 | **84.59** | 77.45 | 77.89 | 79.64 | 80.16 | 82.11 | 83.20 | **84.18** |
| | | 50% | 100% | 86.98 | 87.12 | 87.48 | 87.85 | 89.23 | 89.81 | **90.52** | 87.02 | 87.16 | 87.35 | 82.14 | 89.12 | 89.64 | **90.03** |
| | | 75% | 75% | 81.76 | 81.98 | 82.58 | 83.26 | 85.25 | 86.33 | **87.23** | 81.24 | 81.72 | 82.34 | 83.12 | 85.24 | 86.17 | **87.14** |
| | 70% | 100% | 30% | 76.65 | 76.74 | 76.87 | 76.87 | 79.18 | 80.09 | **81.42** | 76.34 | 76.52 | 77.02 | 77.34 | 78.87 | 79.56 | **81.37** |
| | | 30% | 100% | 85.21 | 86.12 | 86.36 | 86.52 | 87.53 | 88.34 | **88.82** | 84.78 | 85.64 | 85.89 | 86.12 | 87.32 | 88.67 | **89.34** |
| | | 65% | 65% | 79.14 | 79.56 | 80.06 | 81.85 | 82.38 | 82.95 | **83.97** | 78.87 | 79.12 | 79.84 | 81.46 | 81.87 | 82.45 | **83.92** |
| | 90% | 100% | 10% | 72.65 | 73.74 | 73.25 | 74.22 | 75.54 | 76.46 | **78.41** | 71.87 | 73.14 | 73.46 | 74.12 | 75.26 | 76.33 | **78.05** |
| | | 10% | 100% | 82.16 | 82.78 | 83.42 | 84.02 | 86.26 | 87.82 | **89.12** | 81.67 | 82.14 | 83.12 | 83.56 | 85.78 | 86.41 | **87.21** |
| | | 55% | 55% | 77.36 | 77.78 | 78.26 | 78.66 | 80.39 | 81.26 | **82.46** | 76.46 | 76.58 | 77.85 | 78.12 | 79.87 | 81.03 | **82.29** |
| Hateful Memes (AUROC) | 50% | 100% | 50% | 58.32 | 58.56 | 58.78 | 59.31 | 60.24 | 66.41 | **68.96** | 60.56 | 60.31 | 60.87 | 61.87 | 62.32 | 68.25 | **70.82** |
| | | 50% | 100% | 60.34 | 61.12 | 61.34 | 61.78 | 62.34 | 64.93 | **65.79** | 62.41 | 62.35 | 63.13 | 63.88 | 64.46 | 66.80 | **68.45** |
| | | 75% | 75% | 62.34 | 62.87 | 63.14 | 63.24 | 63.78 | 68.09 | **69.63** | 64.87 | 65.84 | 65.46 | 65.86 | 66.02 | 68.16 | **70.81** |
| | 70% | 100% | 30% | 58.54 | 59.02 | 59.36 | 60.02 | 60.56 | 63.70 | **67.52** | 60.74 | 61.12 | 61.26 | 61.56 | 62.82 | 68.94 | **69.53** |
| | | 30% | 100% | 60.12 | 60.78 | 61.32 | 61.54 | 62.32 | 66.79 | **67.77** | 62.74 | 63.24 | 63.14 | 63.48 | 64.12 | 66.98 | **67.59** |
| | | 65% | 65% | 62.34 | 62.56 | 63.12 | 63.32 | 63.78 | 65.57 | **65.89** | 64.82 | 65.04 | 65.23 | 65.48 | 66.08 | 68.42 | **71.08** |
| | 90% | 100% | 10% | 58.02 | 57.34 | 58.32 | 59.02 | 60.34 | 66.23 | **68.16** | 60.03 | 57.21 | 60.74 | 61.14 | 62.08 | 69.70 | **70.96** |
| | | 10% | 100% | 59.02 | 59.32 | 60.21 | 60.56 | 61.34 | 64.66 | **65.16** | 61.46 | 61.52 | 61.87 | 62.42 | 63.87 | 64.54 | **65.57** |
| | | 55% | 55% | 62.32 | 62.56 | 63.24 | 63.78 | 64.34 | 66.74 | **67.21** | 64.32 | 63.34 | 64.85 | 65.37 | 66.78 | 68.03 | **71.23** |

## 4.2. Comparison with the state-of-the-arts

To ascertain the effectiveness of our proposed method, we conduct a comprehensive comparison against state-of-the-art approaches, as detailed in Tab. 1, demonstrates that SPR consistently outperforms the leading SyP (Zhang et al., 2025b) across all missing conditions, exhibiting particular robustness at high missing rates on MM-IMDB (Arevalo et al., 2017). We further observe significant dataset heterogeneity regarding modality dependence. Specifically, datasets reliant on structured semantics like MM-IMDB (Arevalo et al., 2017) and Food101 (Wang et al., 2015) experience severe degradation upon text absence, whereas Hateful Memes (Kiela et al., 2020) shows a drastic decline without visual input. This imbalance highlights the critical need for adaptive prompting mechanisms. While existing methods like SyP (Zhang et al., 2025b) utilize dynamic adapters, they remain limited by coarse granularity and a lack of vector synergy. Conversely, SPR synergistically employs the GIF and LFR modules to capture global and local contextual relationships, combined with the CFS module for fine-grained channel-wise gating. This design allows SPR to precisely capture critical features across diverse missing scenarios, significantly enhancing adaptability to heterogeneous inputs and modal absence.

**Efficiency**. In terms of parameter efficiency, SPR introduces only 1.3M trainable parameters, accounting for merely 0.8%

*Table 2.* Ablation study of SPR under 70% missing rate on both modalities. The best results are highlighted in **bold**.

| GIF | LFR | CFS | Hateful Memes (AUROC) | Food101 (Acc) | MM-IMDB (F1-Macro) |
|---|---|---|---|---|---|
| | | | 64.85 | 79.63 | 50.14 |
| ✓ | | | 66.14 | 81.73 | 52.94 |
| | ✓ | | 67.45 | 82.96 | 53.21 |
| | | ✓ | 65.73 | 82.14 | 51.73 |
| ✓ | ✓ | | 67.92 | 83.13 | 53.82 |
| ✓ | | ✓ | 68.56 | 82.89 | 53.46 |
| | ✓ | ✓ | 68.43 | 83.19 | 54.43 |
| ✓ | ✓ | ✓ | **71.08** | **83.92** | **54.95** |

of the 151M full model and representing only 32% of the parameters required by the DCP (Shi et al., 2024) method. Nevertheless, our method substantially outperforms DCP (Shi et al., 2024) across all missing-modality scenarios. This demonstrates that under conditions of modal absence, how a prompt is refined is more critical than how it is generated.

## 4.3. Ablation study

**Contribution Analysis of SPR Core Components**. We conduct various ablation experiments to evaluate the impact of each component within SPR under a 70% both missing

*Table 3.* Ablation study of Key Design Choices under 70% missing rate on both modalities. The best results are highlighted in **bold**.

| Module | Variant | HateMemes (AUROC) | Food101 (Acc) | MM-IMDb (F1-Macro) |
|---|---|---|---|---|
| | only prompt | 64.85 | 79.63 | 50.14 |
| **GIF** | MLP | 65.12 | 80.47 | 51.32 |
| | DW-Conv | 65.37 | 81.26 | 52.28 |
| | Attention | 63.95 | 78.13 | 49.23 |
| | w/o Residual | 63.13 | 78.59 | 48.95 |
| | **All GIF** | **66.14** | **81.73** | **52.94** |
| **LFR** | MLP | 65.76 | 80.93 | 50.82 |
| | BiGRU | 66.23 | 81.26 | 51.23 |
| | Attention | 66.49 | 81.43 | 52.54 |
| | w/o Residual | 63.27 | 77.91 | 49.27 |
| | **All LFR** | **67.45** | **82.96** | **53.21** |
| **CFS** | MLP | 65.49 | 81.29 | 51.26 |
| | w/o GLU | 64.93 | 80.12 | 50.75 |
| | **All CFS** | **65.73** | **82.14** | **51.73** |

case and summarize the results in Tab. 2. The experimental results clearly demonstrate that incorporating any single module independently yields significant performance improvements, proving their individual contributions. Furthermore, these modules exhibit strong complementarity, as their pairwise combinations achieve performance gains exceeding those of single modules. Ultimately, the complete SPR model, integrating all three modules, achieved the best performance across all three datasets. This robustly validates that SPR, by synergistically refining the prompt's internal structure from global, local, and channel dimensions, can most effectively generate robust guidance signals to address the severe challenge of missing modalities.

**Ablation on Key Design Choices**. The comprehensive ablation study in As shown in Tab. 3 rigorously validates the key designs of our SPR network. The GIF module's core advantage stems from its use of a Bi-GRU to capture complex cross-layer long-range dependencies, which MLP or Depth-wise Conv cannot adequately model; conversely, attention mechanisms falter due to the inherent shortness of the inter-layer sequences. The ablation study for the CFS module confirms that its performance gain is strictly attributed to the dynamic gating mechanism. This mechanism facilitates adaptive channel suppression and enhancement tailored to specific missing modalities. In contrast, using a simple MLP for mapping or scoring yields suboptimal results. Critically, ablating residual connections in the GIF and LFR modules triggers catastrophic performance degradation, affirming the indispensability of residual learning under high missing-rate conditions with scarce supervision. Consequently, the residual pathway functions not merely as a stabilizer, but as the pivotal mechanism enabling the

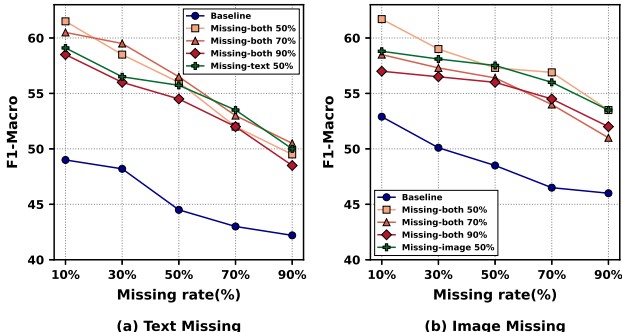

*Figure 3.* Generalizability analysis across varying missing rates on the MM-IMDb validation set. (a) Evaluation on missing-text samples comparing models trained under missing-both versus missing-text settings. (b) Evaluation on missing-image samples comparing models trained under missing-both versus missing-image settings.

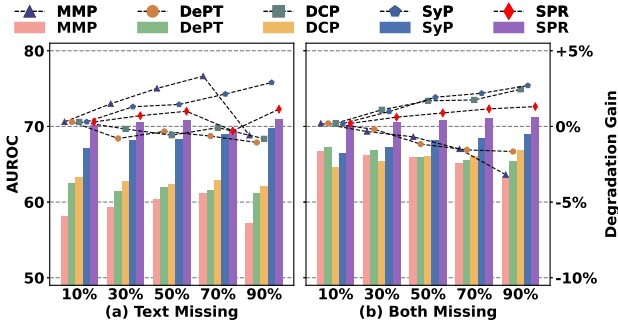

*Figure 4.* Robustness analysis on the HateMemes dataset across various missing rates in terms of AUROC. (a) Evaluation on missing-text samples comparing the proposed SPR with other baseline methods. (b) Evaluation on missing-both samples comparing the proposed SPR with other baseline methods.

modules to operate as fine-grained structural rectifiers.

**Generalizability to Different Missing Rates**. To verify the robustness of SPR against unseen missing patterns, we evaluate the generalizability of models trained under the missing-both setting when applied to single-modality missing test cases. As illustrated in Fig. 3, we present the F1-Macro scores on the MM-IMDb validation set across varying missing rates from 10% to 90%. Two key observations can be drawn from the results. First, SPR consistently outperforms the Baseline method by a significant margin across all missing rates in both missing-text and missing-image scenarios, as shown in Fig. 3(a) and Fig. 3(b) respectively. Notably, even under extreme conditions such as a 90% missing rate, SPR maintains a relatively stable performance, whereas the Baseline exhibits a sharp degradation. This confirms that our structured prompt refinement strategy effectively mitigates the representational collapse caused by severe data incompleteness. Second, models trained under the missing-both setting, represented by the orange and red lines, demonstrate strong zero-shot generalizability to specific single-modality

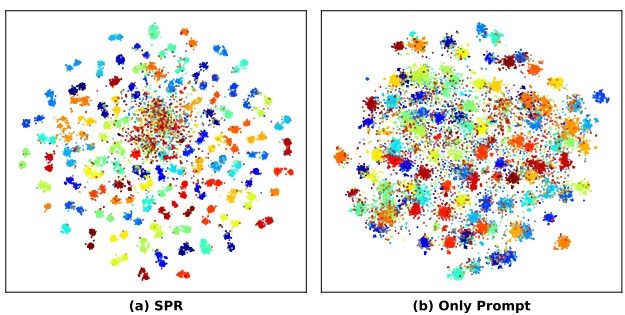

*Figure 5.* t-SNE visualization of SPR and Only Prompt on the Food101 dataset under a 70% both missing rate.

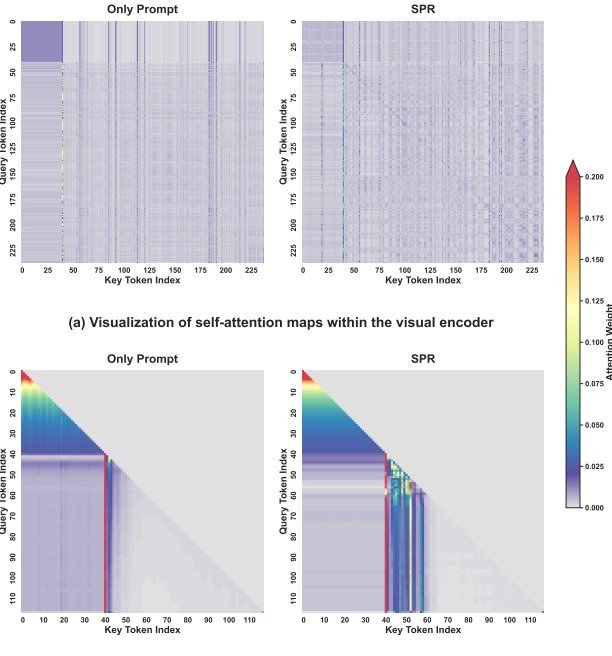

*Figure 6.* Visualization of self-attention maps within the visual and text encoders on the Food101 dataset under the Both-Missing scenario with a 50% missing rate.

missing scenarios. As shown in the figures, the performance curves of models trained on missing-both closely align with those explicitly trained on the target missing modality, such as the Missing-text 50% case. This indicates that SPR learns a generalized, modality-agnostic structural representation that is robust to diverse missing patterns, eliminating the need for retraining specific models for every potential missing scenario.

**Robustness to Different Missing Rates**. To evaluate the robustness of our proposed SPR against varying missing rates, we conducted extensive experiments, with the results illustrated in Fig. 4. We observe that as the modality missing rate escalates from 10% to 90%, all baseline models suffer from significant performance degradation. In stark contrast,

SPR consistently outperforms all baselines across all missing rates. It exhibits minimal performance fluctuations, with its degradation curve remaining stable near the 0% line. Remarkably, in scenarios involving text-missing and both-missing data at high missing rates, SPR even demonstrates performance gains. These results underscores the value of SPR's key components, which are tailored to mitigate the impact of missing modalities. Specifically, by leveraging the unique Structured Prompt Refining network, SPR generates prompt vectors characterized by high semantic coherence and noise resistance. This enables stable and efficient guidance for the frozen backbone model, even under severe missing modality conditions.

### 4.4. Visualization

We employ t-SNE visualization on the Food101 dataset with a 70% missing rate to evaluate prompt geometry, as shown in Fig. 5. While baseline prompts exhibit chaotic distributions with heavily entangled class boundaries (Fig. 5(b)), indicating a failure to maintain semantic coherence, SPR induces a highly organized manifold characterized by distinct, compact clusters (Fig. 5(a)). This enhanced separability validates the efficacy of the synergistic GIF, LFR, and CFS modules, which explicitly encode structural priors to enable the extraction of discriminative features despite severe data incompleteness.

By visualizing the self-attention maps, we provide insight into the mechanism of SPR. As illustrated in Fig. 6(a), within the visual encoder, the baseline exhibits pronounced attention collapse and low-rank vertical stripes stemming from a lack of structural guidance; in stark contrast, SPR successfully reshapes interaction patterns through the synergistic integration of the Global Interaction Fusion, Local Feature Refinement, and Channel Feature Selection modules, restoring the diagonal local dependency structures of image tokens and the grid-like semantic interactions among prompt vectors. As shown in Fig. 6(b) for the text encoder, the self-attention maps exhibit a distinct lower triangular pattern, attributable to the causal masking mechanism in the text encoder. Under this topological constraint, the baseline displays a diffuse and vertical distribution of attention weights, suggesting that unstructured prompts function as isolated tokens with limited internal connectivity. In contrast, SPR manifests a prominent checkerboard texture with significantly higher attention intensity. This structured pattern indicates that SPR successfully establishes strong local contextual dependencies among prompt vectors, evolving them from discrete tokens into coherent.

### 5. Conclusion

In this work, we proposed SPR, a novel Structured Prompt Refining network to address the missing-modality issue by

explicitly refining the internal structure of prompt vectors. This parameter-efficient network refines the three dimensions of the prompt vectors through three distinct modules. Extensive experiments on three mainstream datasets demonstrate SPR's superiority and robustness in tackling incomplete modality learning, achieving state-of-the-art results even at high missing rate.

## Acknowledgements

This work was supported by the National Science Fund for Distinguished Young Scholars of China (Grant No. 62325601), Beijing Municipal Natural Science Foundation (Grant No. L247011), the National Science Foundation for Young Scientists of China (Grant No. 62306300), and the Fundamental and Interdisciplinary Disciplines Breakthrough Plan of the Ministry of Education of China (Grant No. JYB2025XDXM612).

## Impact Statement

This paper presents work whose goal is to advance the field of Machine Learning. There are many potential societal consequences of our work, none which we feel must be specifically highlighted here.

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

# A. Appendix.

## A.1. Hyper-Parameter Analysis

**Impact of Prompt Depth** As illustrated in Fig. 7, our ablation analysis on the MM-IMDb dataset (Arevalo et al., 2017) regarding prompt insertion depth ranging from layer 1 to 12 reveals that model performance follows a distinct trajectory of initial improvement followed by a decline and culminates in optimal effectiveness across various missing modality scenarios when prompts are introduced at the sixth layer. These findings indicate that while shallow prompt insertion fails to offer adequate semantic guidance to rectify deviations arising from missing modalities, excessively deep insertion into the higher layers compromises the semantic integrity of the pre-trained backbone and induces negative transfer, which justifies our decision to configure the prompt depth at six layers to strike an optimal balance between enhancing feature adaptability and preserving pre-trained knowledge.

## Impact of Prompt Length

We conducted a sensitivity analysis on the prompt length hyperparameter by varying the number of learnable tokens from 6 to 64 on the MM-IMDb dataset (Arevalo et al., 2017) under a 70% missing rate condition. As illustrated in Fig. 8, the model performance exhibits an inverted U-shaped trend where the F1-Macro score initially improves as the sequence length increases, indicating that a sufficient number of tokens is critical for encoding the complex structural priors required to compensate for missing modalities. Specifically, the performance for the Both Missing and Image Only settings culminates at a length of 40, while the Text Only setting maintains robust performance at this level after peaking slightly earlier at 36. Conversely, extending the sequence beyond 40 tokens leads to a consistent degradation in performance across all scenarios, which we attribute to the increased optimization difficulty and the introduction of redundant noise that disrupts the efficient guidance of the frozen backbone. These empirical findings substantiate our default configuration of a length of 40, as it strikes an optimal balance between maximizing the representational capacity for feature refinement and minimizing the risk of overfitting or negative transfer associated with excessively long prompts.

**Impact of Kernel Size in Local Refinement** We present a granular sensitivity analysis of the depthwise convolution kernel size $c$ within the Local Feature Refinement module, as this hyperparameter fundamentally dictates the span of the receptive field for local context aggregation. As evidenced by the performance trends in Fig. 9, a distinct inverted U-shaped trajectory emerges across diverse datasets and missing modality scenarios, highlighting a critical trade-off between contextual sufficiency and semantic purity. In

particular, employing a narrow kernel size such as 3 severely restricts the receptive field to immediate neighbors. This spatial myopia prevents the effective capture of broader contextual dependencies, leaving prompt tokens largely isolated and unable to form the cohesive local structures necessary for semantic reinforcement. On the other extreme, expanding the kernel size beyond 11 introduces a detrimental effect where the aggregation scope becomes excessively broad. This over-expansion leads to the assimilation of noise from distant and semantically irrelevant tokens, resulting in feature over-smoothing and a dilution of the intrinsic distinctiveness of individual prompt vectors. Such interference disrupts the delicate process of evolving discrete prompt tokens into structured semantic units, thereby undermining the semantic consistency required to compensate for absent data. Consequently, this structural degradation significantly diminishes the efficacy of the refined prompts in guiding the frozen backbone, preventing the model from achieving optimal robustness under challenging missing modality conditions.

*Table 4.* Quantitative results with ViLT as our backbone on the MM-IMDb, UPMC Food-101, and Hateful Memes datasets with missing rate $\eta = 70\%$ under various modality-missing scenarios. The bold number indicates the best performance.

| Datasets | Image | Text | only prompt | Ours |
|---|---|---|---|---|
| MM-IMDb (F1-Macro) | 100% | 30% | 42.29 | **51.12** |
| | 30% | 100% | 46.30 | **53.29** |
| | 65% | 65% | 45.24 | **52.17** |
| Food101 (Accuracy) | 100% | 30% | 73.62 | **80.28** |
| | 30% | 100% | 84.26 | **88.64** |
| | 65% | 65% | 78.24 | **82.15** |
| Hateful Memes (AUROC) | 100% | 30% | 61.51 | **67.75** |
| | 30% | 100% | 62.38 | **66.14** |
| | 65% | 65% | 64.97 | **69.23** |

## A.2. Deployment on other backbones

To verify the flexibility of our method, we deploy it upon ViLT, a single-stream backbone that processes concatenated text and image features via a common transformer. We compare our approach against the only prompt baseline, which simply incorporates four basic prompt vectors into the ViLT architecture without any structural refinement. Quantitative results in Tab. 4 show that under a 70% missing rate, our method consistently outperforms this unrefined baseline across the MM-IMDb, UPMC Food-101, and Hateful Memes datasets. This superiority demonstrates the robustness of our approach in various missing scenarios. Furthermore, we observe that employing CLIP as the backbone yields superior performance compared to ViLT, which we attribute to the stronger feature representation capabilities

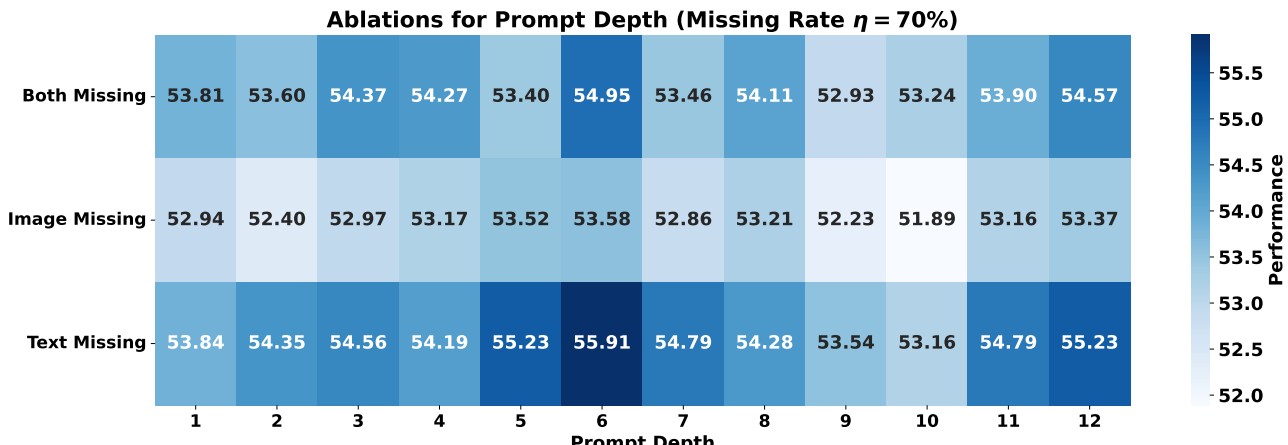

*Figure 7.* Ablations for the prompt depth on the MM-IMDB dataset with missing rate $\eta = 70\%$.

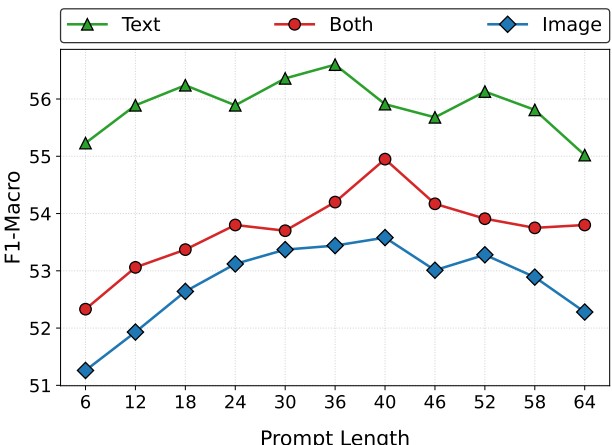

*Figure 8.* Ablations for the prompt length on the MM-IMDB dataset with missing rate $\eta = 70\%$.

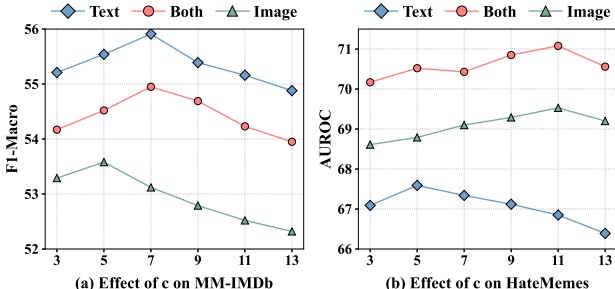

*Figure 9.* Hyper-parameter analysis of depthwise convolution kernel size $c$ under three missing modality scenarios with a 70% missing rate.

derived from CLIP's large-scale contrastive pre-training.

### A.3. Generalizability to Different Missing Rates

To rigorous evaluate the cross-scenario generalization capability on the Hate Memes dataset we conduct a comparative study between models trained under a composite missing-both setting and those explicitly optimized for single-modality missingness as visualized in Fig. 10. The experimental results reveal a substantial performance superiority where the proposed framework consistently outperforms the baseline across the entire spectrum of missing rates ranging from 10% to 90% while exhibiting a significantly more stable degradation trajectory. A critical observation from the subplots is that the performance curve of the model trained on the comprehensive missing-both data closely mirrors that of the specialized models trained specifically for text-missing or image-missing scenarios. This

tight alignment provides strong empirical evidence that the learned structural representations possess a high degree of universality allowing a single unified model to effectively adapt to diverse and unseen missing patterns without the necessity for scenario-specific retraining.

We further extend the generalizability analysis to the Food101 dataset to verify the robustness of the framework under a different data distribution as illustrated in Fig. 11. The visualization confirms that the proposed method maintains a significant performance lead over the baseline approach which suffers from severe deterioration as the missing rate increases towards 90%. Crucially the empirical evidence demonstrates that the model trained under the generic missing-both protocol achieves performance levels nearly identical to those of models fine-tuned for specific missing-text or missing-image conditions. This observation reinforces the conclusion that the global and local refinement mechanisms within the network successfully capture modality-agnostic structural priors thereby enabling the deployment of a single robust model capable of adapting to arbitrary modality availability during inference while elim-

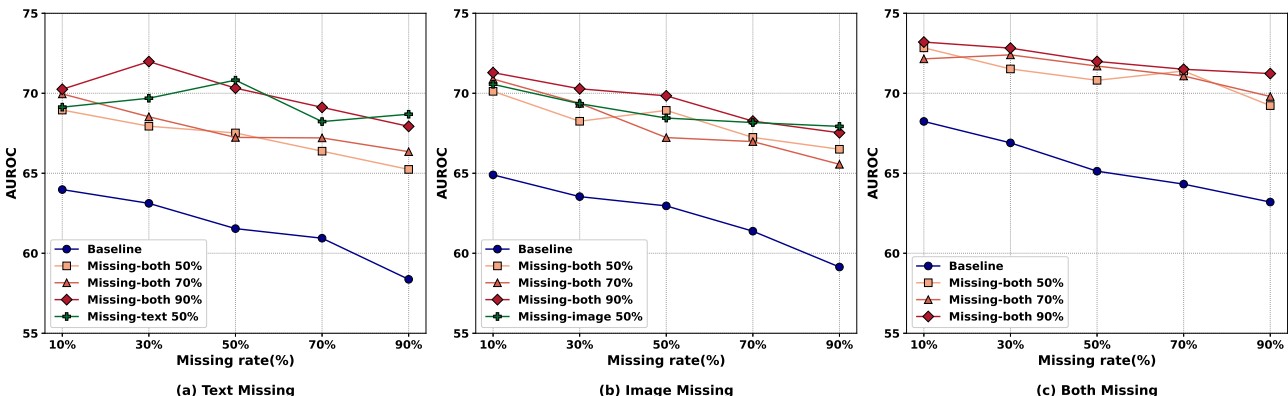

*Figure 10.* Ablations on the generalizability to different testing scenarios across various missing rates on the val set of Hate Memes dataset. (a) Models are trained on missing-both or missing-text cases, and evaluated on missing-text cases with different missing rates. (b) Models are trained on missing-both or missing-image cases, and evaluated on missing-image cases with different missing rates. (c) All models are trained on missing-both cases, and evaluated on missing-both cases with different missing rates.

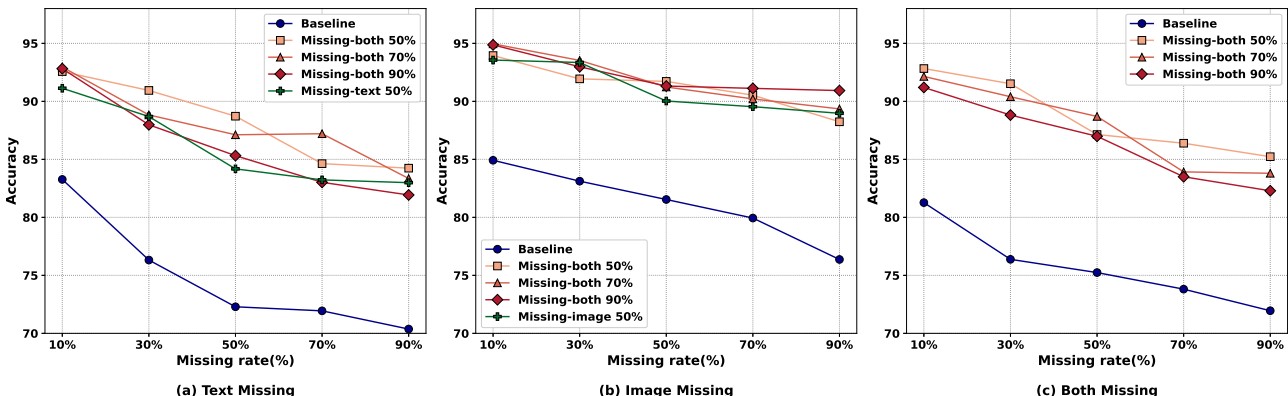

*Figure 11.* Ablations on the generalizability to different testing scenarios across various missing rates on the val set of Food101 dataset. (a) Models are trained on missing-both or missing-text cases, and evaluated on missing-text cases with different missing rates. (b) Models are trained on missing-both or missing-image cases, and evaluated on missing-image cases with different missing rates. (c) All models are trained on missing-both cases, and evaluated on missing-both cases with different missing rates.

inating the computational burden of maintaining multiple specialized models.

## A.4. Compatibility with complete data

To investigate the robustness of our framework against dynamic data unavailability, we evaluated the models across a continuous spectrum of missing rates ranging from 0% to 100% in the Missing-Both scenario. As depicted in Fig. 12, the proposed SPR network consistently surpasses both the Baseline and the Only Prompt variants by a substantial margin. We observe that while the Baseline suffers from a sharp performance decline as the missing rate increases, SPR exhibits a trajectory of graceful degradation and maintains high categorization accuracy even under extreme sparsity. This resilience is attributed to the synergistic effect of the Global Interaction Fusion and Channel Feature Selection modules, which actively recalibrate the prompt represen-

tations to preserve semantic integrity despite the varying severity of modality absence.

## A.5. Model Generalizability

To comprehensively evaluate the generalization capability of SPR against state-of-the-art approaches we conducted a comparative analysis on the MM-IMDb dataset under varying missing rates of 50%, 70%, and 90%. As illustrated in Fig. 13 the proposed SPR consistently outperforms all baseline methods across all tested configurations which is visually represented by the highest color intensity in the heatmaps. This superiority becomes particularly pronounced in the challenging both-missing scenario at an extreme 90% missing rate where SPR achieves an F1-Macro score of 53.41% and surpasses the leading competitor SyP by a substantial margin of 3.78% while exceeding static methods like CoOp by over 9%. These empirical results

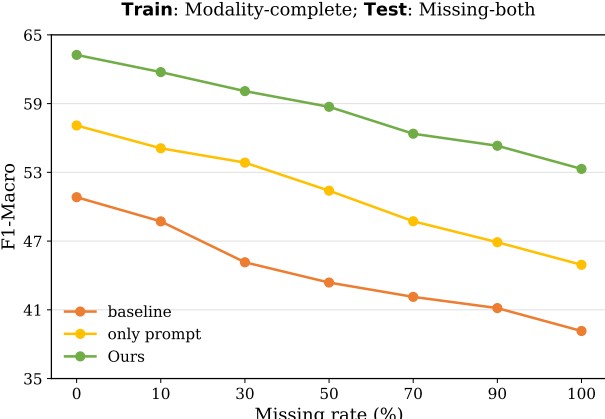

*Figure 12.* All models are trained using modality-complete data. To simulate potential missing modality scenarios during the testing phase, we randomly assign each data pair to a specific modality configuration such as text-only, image-only, or modality-complete across different training epochs. Our evaluation focuses on the missing-both scenario under varying missing rates.

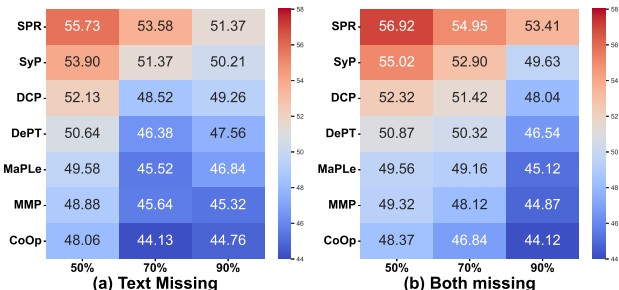

*Figure 13.* Generalization analysis on the MM-IMDb (Arevalo et al., 2017) dataset across various missing rates in terms of F1_Macro

robustly validate that the structured refinement strategy of SPR does not merely rely on pre-trained knowledge but actively reconstructs missing semantics to prevent representational collapse and ensure stable generalization even in environments with severe data sparsity.

