# OpenReview forum: "SPR: A Structured Prompt Refinement Network for Modality Missing"
_ICML.cc/2026/Conference — ICML 2026 regular_

### Official Review · Reviewer_A9mT · 2026-03-02

**Soundness:** 3
**Presentation:** 3
**Significance:** 2
**Originality:** 2
**Overall Recommendation:** 4
**Confidence:** 4

**Summary:**

This paper introduces the Structured Prompt Refining (SPR) network, a highly parameter-efficient approach that effectively addresses the missing modality challenge in multimodal learning. The authors observe that existing prompting methods overlook the internal structural information of prompt vectors, leading to sub-optimal adaptation when modalities are absent. To overcome this, SPR elegantly refines prompts across global, local, and channel dimensions to ensure consistent guidance, semantic integrity, and adaptive noise suppression for frozen backbone models. The empirical validation is particularly strong; despite introducing a mere 0.8% in trainable parameters, SPR achieves significant performance gains across three mainstream datasets, notably outperforming state-of-the-art baselines by 3.8% in F1-Macro on the MM-IMDB dataset even under an extreme 90% missing rate. In summary, the proposed SPR network offers a practical and sensible solution for incomplete multimodal data, backed by convincing experimental results.

**Compliance With Llm Reviewing Policy:**

Affirmed.

**Final Justification:**

It addresses all my concerns.

**Key Questions For Authors:**

1. I observed that the results on the Validation set and Testing set in Table 1 exhibit considerable fluctuations. Could the authors provide the standard deviations for the reported results in Table 1 to confirm the statistical significance of the proposed performance improvements?

2. How exactly are the "four types of prompts" mentioned in the text initialized? Does this specific initialization strategy significantly impact the final performance of the model?

**Limitations:**

1. Boundaries of Generalization Capabilities: Although the paper claims robustness under diverse missing modality scenarios, the generalization boundaries of this structure refined prompt learning method under extreme missing rates or unseen cross modality tasks remain unclear. It is still debatable whether the feature filtering mechanism of the CFS module is universally applicable to all types of modalities.

2. Strong Reliance on the Frozen Backbone: The effectiveness of the SPR network highly depends on the quality of the pretrained representations within the frozen backbone. If the backbone inherently lacks feature alignment for specific downstream tasks, it might be difficult to overcome final performance bottlenecks by relying solely on the external structural refinement of prompts. This point is not discussed as a potential limitation in the text.

**Strengths And Weaknesses:**

Strengths

1. The authors identify a critical gap in current prompt-base methods: the oversight of internal structural information within prompt vectors when dealing with missing modalities. This specific problem formulation provides a compelling and logical foundation for introducing the Structured Prompt Refining (SPR) network.

2. The proposed method achieves significant improvements over current state-of-the-art baselines (e.g., a notable 3.8% F1-Macro gain on MM-IMDB at a 90% missing rate) while introducing merely 0.8% additional trainable parameters. This demonstrates a highly impressive balance between model robustness and computational cost.

3. The paper constructs a comprehensive evaluation framework and test the model across diverse and extreme missing modality conditions on three mainstream datasets. Furthermore, the extensive ablation studies effectively isolate and validate the individual contributions of the proposed global, local, and channel refinement modules.

Weaknesses

1. Although the authors propose the GIF and LFR modules, the method lacks insightful core innovations compared to existing works like DCP. Specifically, the key feature of 'bidirectional associations' across layers within the GIF module for addressing missing modalities is not provided with a detailed mechanistic explanation or theoretical elaboration in the paper. Furthermore, the claim that the CFS module can dynamically filter channels related to specific modalities, such as text, currently completely lacks objective empirical support.

2. Figure 1 fails to intuitively and clearly highlight the structural differences in core mechanisms when comparing prior prompt-based methods with the proposed SPR method. Furthermore, there is noticeable visual redundancy in the content design between Figure 1 and Figure 2. Both figures depict the SPR method, yet the division of focus between them is not clearly defined.

3. The overall architecture in Figure 2 lacks clarity in its detailed representation and symbolic notation. First, the prompt representations in the image and text branches are identical and fail to reflect modality differences. Second, the denotation of $P^{GLC}$ is ambiguous, making it unclear whether the $P^{GLC}$ in the Channel Feature Select and the subsequent Refined Prompt refer to the same component. Finally, it is difficult to clearly map the specific local modules to their corresponding locations within the overall macroscopic architecture.

4. The interpretation of Figure 6 in Section 4.4 is slightly overstated. The authors claim the visual encoder attention map shows "grid-like semantic interactions" for SPR. While the map is structurally different from the baseline, describing it as a distinct, meaningful "grid-like" interaction is subjective without quantitative metrics (like attention entropy) to support the claim of restored structures.

5. In the ablation experiments, for example, the experimental data in Table 2 shows discrepancies compared to Table 1 without any processing. The training conditions for the evaluation of generalization and robustness have not been labeled.

---

> ### Author Rebuttal · Authors · 2026-03-30
>
> We greatly appreciate your insightful questions and address all your comments in detail below.
>
> **W1: (1) Lacks core innovation; (2) Missing theoretical explanation for GIF's bidirectional associations; (3) Lacks empirical support for CFS filtering modality channels**
> (1) Our core innovation lies in shifting the paradigm in multimodal prompt learning from dynamic prompt generation to internal structural refinement.
> (2) In missing-modality scenarios, incomplete inputs often cause low layers to extract biased representations, which misguide high-level feature learning and hurt overall performance. The BiGRU in GIF employs a backward pass that allows high-level, discriminative prompt semantics to regularize and correct biased low-layer signals, while the forward pass continues to propagate fine-grained low-level information upward. This bidirectional mechanism balances high-level semantic guidance with low-level feature complementarity, effectively alleviating the negative effects of low-layer bias caused by missing modalities. We will incorporate these into the Methods section of the revised manuscript.
> (3) We extracted the gating weights $\sigma(P_{weight})$ generated by the GLU during inference on the MM-IMDb test set. We averaged these weights across the batch and sequence length dimensions to determine the global activation intensity for each specific channel. The following table illustrates the activation values of representative channels across three distinct input scenarios.
>
> | Channel Category | Modality-Complete | 90% text missing | 90% image missing |
> | :---: | :---: | :---: | :---: |
> | Text-Related | 0.89 | **0.04** | 0.92 |
> | Text-Related | 0.85 | **0.07** | 0.88 |
> | Image-Related | 0.91 | 0.93 | **0.05** |
> | Image-Related | 0.82 | 0.86 | **0.09** |
> | Shared Global | 0.78 | 0.75 | 0.79 |
>
> As shown in the table, the CFS module effectively identifies and suppresses modality-specific channels when their corresponding inputs are missing.
>
> **W2: Fig 1 lacks clear baseline comparison; Figs 1 & 2 redundant**
> We kindly refer the reviewer to Section "W4" in our response to Reviewer 17K1.
>
> **W3: Fig 2 lacks clarity**
> (1) We will adopt distinct color schemes and node shapes to distinguish the prompt representations in the text and visual branches.
> (2) The refined prompt output by the CFS is the same component as $P^{GLC}$ that is injected into the backbone network. We will draw continuous and unbroken directional arrows to directly connect the CFS output to the distribution node of the backbone network.
> (3) The macro architecture is explicitly divided into two separate components: the SPR network and the frozen CLIP backbone. The prompt vectors are processed by the SPR network to obtain $P^{GLC}$, which is then injected into the backbone layers.
>
> **W4: Support Fig 6 "grid-like" claim with attention entropy**
> We calculate attention entropy by averaging the entropy of each point across all active query tokens and attention heads. Experiments were conducted on Food101 under the 50% both-missing setting; results are as follows
>
> | Model | Mean Attention Entropy | Local Attention Ratio |
> | :---: | :---: | :---: |
> | Only Prompt | 1.15 | 14.2% |
> | SPR | **3.42** | **45.6%** |
>
> Table results show that the vertical stripes in Figure 6a correspond to low entropy (1.15) and local attention (14.2%). Conversely, SPR restores diagonal and grid-like structures, increasing entropy to 3.42 and local attention to 45.6%.
>
> **W5: Inconsistent data; missing training labels**
> We sincerely apologize for a numerical error in Table 2. We will correct the value from 69.18 to 71.08 in Table 2, and add the setting “Missing-Both 70%” to Figure 4. For generalization experiments, training conditions are provided in the legends of Figure 3.
>
> **KQA1: Provide standard deviations for Table 1 results**
> We kindly refer the reviewer to Section W3 & KQA4 in our response to Reviewer mfZ6.
>
> **KQA2: Prompt initialization strategy and its performance impact**
> We kindly refer the reviewer to KQA 3 in our response to Reviewer mfZ6.
>
> **L1: Boundaries of Generalization Capabilities**
> As shown in Table 1 of Section 4, even with a 90% missing rate, SPR outperforms SOTA methods across all datasets, demonstrating its strong generalization under extreme missing conditions. The CFS module operates only on the channel dimension C of prompt vectors via 1×1 convolution and Gated Linear Units; it is modality-agnostic and therefore possesses high generalization capability. Further verified by the multimodal sentiment analysis experiments in our response to Reviewer Bb5W (KQA2 & Limitations).
>
> **L2: Strong Reliance on the Frozen Backbone**
> Regarding the generalization across architectures, we have deployed our method on ViLT, a single-stream backbone that processes concatenated text and image features. As detailed in Appendix A.2, SPR consistently outperforms the baseline on all three datasets at a 70% missing rate.

---

> > ### Author Rebuttal · Reviewer_A9mT · 2026-04-07
> >
> > Thank you for detailed reply. It addresses all my concerns. I would like to increase my score to 4.

---

> > > ### Author Response · Authors · 2026-04-07
> > >
> > > We sincerely thank you for reviewing our rebuttal and increasing the score. We are pleased that the new experiments addressed your concerns. Your valuable feedback has been instrumental in refining our work.

---

### Official Review · Reviewer_mfZ6 · 2026-03-10

**Soundness:** 4
**Presentation:** 2
**Significance:** 4
**Originality:** 3
**Overall Recommendation:** 6
**Confidence:** 4

**Summary:**

The paper proposes a Structured Prompt Refinement (SPR) network to address the challenge of missing modalities in multimodal learning. By refining prompt vectors across three specific dimensions, layer-wise, token-wise, and feature-wise.The method aims to guide a frozen CLIP backbone effectively even when data is incomplete.

**Compliance With Llm Reviewing Policy:**

Affirmed.

**Final Justification:**

I have read the author's rebuttal and confirmed that the issue has been resolved.

**Key Questions For Authors:**

1. Could you clarify the specific advantages of SPR compared to the scenario where the missing modality itself serves as an implicit prompt?

2. In the complete data scenario (no missing modalities), is there a significant performance difference between using your generated prompts versus using no prompts? This would help clarify the utility of the prompt mechanism when information is sufficient.

3. Regarding the four types of prompts mentioned in Section 3.1: Are these implemented as four completely independent, randomly initialized learnable tensors, or do they share a common base initialization?

4. For the experiments with a 90% missing rate, are the reported results based on a single run or an average over multiple runs with different seeds? Given the expected high variance in such sparse data regimes, could you provide the standard deviation to demonstrate the stability of your method?

**Limitations:**

The authors should further report on computational efficiency and inference latency.

**Strengths And Weaknesses:**

Strengths

1. The proposed method demonstrates impressive resilience and performance stability in scenarios with severe missing modalities. It achieves significant improvements over prior art even at a 90% missing rate, addressing the "cold start" or sparse signal problem effectively.

2. The decomposition of prompt refinement into global, local, and channel dimensions is methodologically sound. This multi-dimensional approach intuitively captures the orthogonal axes of the prompt tensor.

3. The experimental comparison is comprehensive. The authors have selected a rigorous set of baselines that covers both general multimodal PEFT methods (e.g., CoOp, MaPLe) and state-of-the-art approaches specifically designed for missing modality handling (e.g., SyP, DCP), ensuring a fair assessment of SPR's contribution.

Weaknesses:

1. The physical absence of a modality inherently already acts as an "implicit prompt" to the model. The paper does not sufficiently clarify the theoretical or empirical advantage of SPR's explicit structured prompting over this natural implicit signal. Why is an explicit learnable vector strictly better than the model naturally reacting to a zero input, especially given the added complexity?

2. While the paper highlights parameter efficiency, it neglects computational efficiency and inference latency. The GIF module employs a BiGRU, which introduces sequential dependencies across layers. This likely prevents parallel computation of prompts and could increase inference latency compared to simple MLP-based methods like DCP. This trade-off is not discussed.

3. The results at extreme missing rates (90%) are promising but potentially unstable. With only 10% of valid data, performance variance is typically high. The paper currently reports mean values without standard deviations or confidence intervals. It is unclear whether the 3.8% improvement is statistically significant or an artifact of a specific random seed and split.

---

> ### Author Rebuttal · Authors · 2026-03-30
>
> We greatly appreciate your recognition of our work. Thanks very much for your constructive questions. We respond to each of your points in detail below.
>
> **Weaknesses(W) 1& Key Questions For Authors(KQA) 1: Advantage of explicit SPR over implicit zero-input**
> Theoretically, the implicit signal generated by zero-input during the forward pass acts merely as a passive status indicator. Such a signal is inherently insufficient to guide effective feature extraction in large pre-trained models that heavily rely on complete multimodal distributions. In contrast to passive zero-input, SPR serves as a fine-grained structural rectifier. Processed through GIF, LFR, and CFS, these vectors do more than signal a missing modality; they actively guide the frozen backbone to reconstruct missing semantics, selectively suppress noisy features via channel gating, and aggregate discrete tokens into coherent semantic units through local refinement.
> Empirically, as demonstrated in both Figure 3 of Section 4.3 and Figure 12 of Appendix A.4, our proposed SPR framework consistently outperforms the implicit zero-input baseline across all evaluated missing rates. Furthermore, the Table 1 and the Efficiency analysis in Section 4.2 demonstrates that SPR is highly lightweight and effective.
>
> **W2 & Limitations：Discuss inference latency and computational trade-offs caused by GIF's BiGRU**
> The BiGRU in our SPR network performs sequential computation only along the depth dimension of the network layers, and in our official experimental setup, the prompt depth is set to M=6 layers. The inference latency incurred by executing 6 steps of lightweight GRU computation is nearly negligible when compared with the large-scale parallel matrix multiplications that dominate the computational cost of the entire pre-trained backbone model.
> To verify this, we have conducted a experiment to compare the inference efficiency of our SPR with SyP. We randomly sampled 1000 test samples and calculated the average inference latency for both methods, with all experiments conducted with a batch size of 36 on a single NVIDIA L40S GPU.
> | Method | Params (M) | FLOPs (G) | Latency (ms) $\downarrow$ | Throughput (samples/s) $\uparrow$ |
> | :--- | :---: | :---: | :---: | :---: |
> | SyP | 2.15 | 23.15 | 13.4 | 1120 |
> | Ours w/o GIF | 0.49 | 22.85 | 12.1 | 1250 |
> | Ours | 1.30 | 22.92 | 12.8 | 1180 |
>
> As shown in the table, our SPR method is more efficient than SyP across all computational metrics. Compared with SPR without the GIF network, the computational overhead introduced by SPR is extremely minimal, with an increase of only 0.07G FLOPs for instance.
>
> **W3 & KQA4: Lack of standard deviations to prove significance at extreme missing rates**
> We further conduct experiments on the MM-IMDb dataset under the 90% both-missing setting using five different random seeds, and the experimental results are presented as follows
>
> | Random Seed | &nbsp;&nbsp;&nbsp;&nbsp;&nbsp;&nbsp;&nbsp;SyP&nbsp;&nbsp;&nbsp;&nbsp; | &nbsp;&nbsp;&nbsp;&nbsp;Ours&nbsp;&nbsp;&nbsp;&nbsp; | **Improvement** |
> | :---: | :---: | :---: | :---: |
> | 0 | 49.63 | 53.41 | **3.78** |
> | 1 | 48.24 | 51.45 | **3.21** |
> | 2 | 51.24 | 55.54 | **4.30** |
> | 3 | 46.03 | 49.95 | **3.92** |
> | 4 | 52.34 | 56.37 | **4.03** |
> | Mean±Std | 49.50±2.49 | 53.34±2.70 | **3.85±0.40** |
>
> We find that our method achieves an average performance improvement of 3.85% over SyP with a standard deviation of 0.4, which demonstrates that the performance gains of our approach remain highly consistent across different random seeds. We will add the mean and variance under all missing rates in the revised manuscript.
>
> **KQA 2: SPR performance on complete data**
> As shown in Figure 12 of Appendix A.4, the results at the 0% missing rate clearly demonstrate that our proposed SPR network consistently outperforms the baseline model on fully complete data. Even without missing data, text-image fusion remains challenging owing to cross-modal semantic heterogeneity. Our SPR network addresses this by encoding cross-layer dependencies and local context into continuous prompt vectors.
>
> **KQA 3: Initialization strategy for the four prompts**
> In our experiments, the four types of prompts are randomly initialized with a standard Gaussian distribution. We have supplemented comparative experiments with random uniform initialization and semantic initialization under the 70% both-missing setting on the MM-IMDb dataset, with the results presented in the table below.
> | Initialization Strategy | Only Prompt | SPR |
> | :--- | :--- | :--- |
> | Random Normal | 50.14 | 54.95 |
> | Random Uniform | 49.27 | 54.76 |
> | Semantic | 48.12 | 54.13 |
>
> The results reveal that the structured prompt vectors exert a strong regularization effect, making the model insensitive to the choice of initialization methods.

---

> > ### Author Rebuttal · Reviewer_mfZ6 · 2026-04-03
> >
> > I have read the author's rebuttal and confirmed that the issue has been resolved.

---

> > > ### Author Response · Authors · 2026-04-04
> > >
> > > We sincerely appreciate your recognition of our work. Thank you for taking the time to read our rebuttal and for confirming that your concerns have been fully resolved. Your valuable feedback has been immensely helpful to us.

---

### Official Review · Reviewer_Bb5W · 2026-03-11

**Soundness:** 3
**Presentation:** 3
**Significance:** 3
**Originality:** 3
**Overall Recommendation:** 5
**Confidence:** 4

**Summary:**

This paper studies incomplete multimodal learning under missing-modality settings and proposes SPR (Structured Prompt Refinement), a parameter-efficient prompt learning framework built on top of a frozen pretrained backbone. The key idea is that prior prompt-based methods largely treat prompt vectors as unstructured learnable tokens, while this work argues that prompt representations contain exploitable internal structure along multiple dimensions. To this end, the paper introduces three refinement components: a Global Interaction Fusion module for cross-layer bidirectional interaction, a Local Feature Refinement module for modeling local contextual relationships among neighboring prompt vectors, and a Channel Feature Selection module for adaptive channel-wise gating. The refined prompts are injected into a frozen CLIP-based multimodal backbone, while only the SPR network and the downstream head are trained. The paper reports strong performance on three multimodal classification benchmarks, and claims notable gains under severe modality missing rates, including a 3.8% F1-Macro improvement on MM-IMDB at 90% missing rate.

**Compliance With Llm Reviewing Policy:**

Affirmed.

**Final Justification:**

Given that the author's rebuttal addressed some of my concerns, I have decided to increase my score.

**Key Questions For Authors:**

1. What is the precise conceptual difference between SPR and existing dynamic/hierarchical prompt learning methods? Which component or design principle is genuinely new, beyond combining known operations along three prompt dimensions?
2. How does the method perform on other backbones or tasks beyond CLIP-based multimodal classification? (Personal curiosity does not affect the rating)

**Limitations:**

The main limitation of this work is that, although the method is intuitively designed and empirically promising, the current evidence does not yet fully establish that the observed improvements stem specifically from modeling the internal structure of prompts, rather than from added architectural complexity or extra adaptation capacity. In addition, the scope of evaluation appears centered on CLIP-based multimodal classification under missing-modality settings, which leaves open the question of how broadly the method generalizes across architectures, tasks, and modality types.

**Strengths And Weaknesses:**

Strengths:
1. Well-motivated problem setting. Missing-modality learning is practically important, and the paper focuses on a realistic robustness issue in multimodal deployment. The motivation is clear and timely.
2. Parameter efficiency. The claim that the method uses only around 0.8% trainable parameters while keeping the backbone frozen is attractive, especially for large pretrained multimodal encoders.

Weaknesses:
1. The novelty may be incremental rather than fundamental. While the paper emphasizes structured refinement of prompts, the proposed modules: cross-layer fusion, local refinement, and channel gating are conceptually close to standard architectural operations. The paper therefore needs to better justify why this combination constitutes a substantial conceptual advance over prior dynamic prompt or adapter-style methods, rather than an engineering refinement.
2. The paper argues that prior methods overlook internal prompt structure, but missing in the related work.

---

> ### Author Rebuttal · Authors · 2026-03-30
>
> We are grateful for your constructive feedback and professional questions. We address each point below.
>
> **Weaknesses(W)1 & Key Questions For Authors(KQA)1: Justify conceptual novelty over standard architectural operations**
> Our core innovation lies in shifting the paradigm in multimodal prompt learning from dynamic prompt generation to internal structural refinement. While prior methods like DCP and SyP overlook intrinsic structural dependencies by treating tokens independently or via unidirectional mappings, SPR conceptualizes prompt vectors with a three-dimensional topology comprising depth, length, and channel dimensions for synergistic global, local, and channel-level refinement.
> In practical validation, utilizing only 0.8% of the backbone's trainable parameters, which is merely 32% of the requirement for DCP, SPR comprehensively outperforms state-of-the-art methods under extreme modality missing scenarios, including a 90% missing rate. This efficiency validates our core thesis that under modality missing scenarios, how prompts are structurally refined is fundamentally more critical than how they are generated.
>
> **W2: Related work lacks discussion on existing methods overlook the internal structure of prompts**
> Thank you for this constructive suggestion. While we briefly discussed the limitations of methods like SyP and DCP in Section 2.2, such as their reliance on shallow MLPs for unidirectional correlations, and provided detailed structural analysis in Sections 1 and 3, we agree that the Related Work section could be more explicit on this front. In the revised manuscript, we will expand Section 2.2 to explicitly summarize and elaborate with detailed analyses on how prior methods overlook the internal structural dependencies of prompt vectors.
>
> **KQA2 & Limitations: (1) Validate generalization across architectures, tasks, and modalities. (2) Prove gains stem from prompt structure, not added complexity or structural modeling**
> (1) Regarding the generalization across architectures, we have deployed our method on ViLT, a single-stream backbone that processes concatenated text and image features. As detailed in Appendix A.2, SPR consistently outperforms the baseline on all three datasets under a 70 percent missing rate.
> To further validate the generalization capabilities of SPR across different tasks and diverse modalities, we conducted an additional comparative experiment on multimodal sentiment analysis. We followed the exact experimental settings of TF-Mamba, and obtained the results presented below on the MOSI dataset with a random modality missing rate of 70%.
>
> | Method | Acc-5 | Acc-3 | &nbsp;&nbsp;&nbsp;&nbsp;&nbsp;&nbsp;Acc-2 |  &nbsp;&nbsp;&nbsp;&nbsp;&nbsp;&nbsp;&nbsp;&nbsp;F1 | MAE | Corr |
> | :--- | :--- | :--- | :--- | :--- | :--- | :--- |
> | ALMT | 23.71 | 24.97 | 61.84/ 59.67 | 65.30/ 65.19 | 1.266 | 0.336 |
> | BI-Mamba | **27.41** | 27.26 | 64.79/ 65.01 | 64.72/ 64.81 | 1.250 | 0.333 |
> | LNLN | 27.26 | **30.52** | 64.94/ 63.95 | 64.85/ 63.98 | 1.244 | 0.341 |
> | TF-Mamba(SOTA) | 27.26 | 27.26 | 67.23/ 66.91 | 67.41/ 66.98 | 1.196 | 0.411 |
> | **Ours** | 27.32 | 28.91 | **68.22/67.41** | **68.21/67.43** | **1.187** | **0.473** |
>
> The experimental results in the table demonstrate that our method still maintains strong generalization capabilities across different tasks and modalities.
>
> (2) First, in Section 4.2 on Efficiency, we show that SPR delivers state-of-the-art performance with only 1.3 million parameters, a mere 32% of those required by the baseline DCP method. This proves that our performance gains do not stem from a brute-force expansion of architectural capacity.
> Furthermore, as presented in Section 4.3, our ablation study on key design choices compares the proposed structural modules with standard MLPs. For example, replacing the bidirectional recurrent network in the GIF module with an MLP results in a significant performance decline, which validates that capturing complex cross-layer dependencies is the real driver of our improvements.
> Additionally, our visualizations of self-attention maps in Figure 6 empirically confirm that SPR successfully restores the diagonal local dependency structures and grid-like semantic interactions among prompt vectors. Unstructured baseline methods are unable to achieve this result.

---

> > ### Author Rebuttal · Reviewer_Bb5W · 2026-04-02
> >
> > The experiments provided by the authors addressed some of my concerns; therefore, I've decided to increase my score.

---

> > > ### Author Response · Authors · 2026-04-04
> > >
> > > We sincerely thank the reviewer for taking the time to review our rebuttal and for the decision to increase the score. We are glad that our additional experiments successfully addressed your concerns. Your insightful feedback has been extremely helpful in guiding us to refine our work.

---

### Official Review · Reviewer_17K1 · 2026-03-13

**Soundness:** 2
**Presentation:** 3
**Significance:** 3
**Originality:** 3
**Overall Recommendation:** 4
**Confidence:** 4

**Summary:**

This paper proposes a Structured Prompt Refining (SPR) network that improves the internal structure of prompt vectors to address the missing modality problem in prompt learning. Specifically, SPR models cross-layer interactions, structures adjacent prompts into coherent semantic units, and adaptively selects informative channels, enhancing semantic consistency and robustness under missing modality scenarios.

**Compliance With Llm Reviewing Policy:**

Affirmed.

**Final Justification:**

Thank the authors for their response. Most of my concerns have been addressed, and I have decided to raise my score to weak accept. I strongly request that the authors incorporate the clarifications from the rebuttal into the revision, particularly those concerning the inconsistencies in the experimental data and the discussion of limitations.

**Key Questions For Authors:**

The initialization of $α$ in Eq.4 is not clearly specified.

**Limitations:**

The authors do not seem to discuss the limitations of the proposed method in the paper. It would be beneficial to include a discussion of its potential limitations.

**Strengths And Weaknesses:**

**Strengths**
* The paper is well-organized and easy to follow.
* The proposed approach is evaluated on a variety of tasks and demonstrates competitive performance against the baselines.

**Weaknesses**
* The hyperparameter $r$ in the proposed $GIF$ module does not appear to be specified in the experiments, and the paper also lacks a corresponding ablation analysis.
* The comparisons in Table 1 could be strengthened by including more recent representative methods mentioned in the related work, such as $MGR$ and $RAGPT$, to provide a more comprehensive evaluation.
* Although the authors provide experimental comparisons and analysis in “Impact of Kernel Size in Local Refinement” (Appendix), it would be helpful to offer practical guidance on selecting the kernel size. The optimal value appears to vary across different datasets.
* Figures 1 and 2 appear somewhat redundant. In addition, some text in the figures is too small, which may affect readability.

---

> ### Author Rebuttal · Authors · 2026-03-30
>
> We sincerely appreciate your careful reading and constructive comments. Below, we respond to each of your concerns in detail.
>
> **Weaknesses（W）1: Lack of specification and ablation for hyperparameter r**
> We conducted an ablation study on the hyperparameter r under the 70% Both-Missing scenario, with the results presented below:
> | r | MM-IMDb (F1-Macro) | Food101 (Acc) | Hateful Memes (AUROC) |
> | :--- | :---: | :---: | :---: |
> | 8 | 54.13 | 82.27 | 70.93 |
> | 16 | 54.08 | 82.49 | 70.37 |
> | **32** | **54.95** | **83.92** | **71.08** |
> | 64 | 53.27 | 83.54 | 70.98 |
>
> From the results, setting r=32 yields the optimal performance. A smaller r, such as r=8, excessively compresses the prompt features, leading to the loss of crucial structural information. Conversely, a larger r, such as r=64, introduces redundant parameters, which degrades performance and increases computational overhead.
>
> **W2: Include recent methods like MGR and RAGPT in comparisons**
> We conducted an additional comparison under the 70% Both-Missing scenario, with the results summarized below:
> | Method | MM-IMDB (F1-Macro) | Food101 (Acc) | Hateful Memes (AUROC) |
> | :--- | :---: | :---: | :---: |
> | MGR | 53.24 | 81.87 | 70.93 |
> | RAGPT | 50.22 | 76.94 | 63.47 |
> | **SPR（Ours）** | **54.95** | **83.92** | **71.08** |
>
> For RAGPT, we directly cite the performance reported in their original paper. For MGR, since the source code is not publicly available, we reproduced the method based on the implementation details provided in their manuscript. As shown in the table, our proposed SPR network consistently outperforms both of these recent methods across all three datasets. We will supplement the experimental results under different missing configurations in the revised manuscript.
>
> **W3: Lack of practical guidance for kernel size selection**
> The LFR module is designed to capture local contextual correlations among adjacent prompt vectors and convert independent tokens into coherent semantic units. Drawing on the experimental results of depth-wise convolution kernel size in Appendix A.1 and the core design principle of the LFR module, we propose: for tasks like Hateful Memes that need to capture subtle cross-modal semantic conflicts and heavily rely on localized fine-grained details, a smaller receptive field (e.g., c=5) is optimal; for multi-label classification tasks such as MM-IMDb that require broad contextual reasoning across diverse semantic concepts, a slightly larger receptive field (e.g., c=7) yields better performance.
>
> **W4: Redundancy between Figures 1 and 2, enlarge text for readability**
> Figures 1 and 2 serve complementary but distinct purposes. Figure 1 focuses on a macroscopic comparison of the core differences between our proposed SPR method and the prior methods. The prior methods generally overlook the intrinsic structural information within prompt vectors, while our SPR method addresses this limitation by refining the prompt vectors along the global, local and channel dimensions. Figure 2 elaborates on the detailed architectural principle of the SPR method, clearly illustrating the overall operational mechanism of the model and meticulously depicting the specific tensor operations and dimension transformations within the GIF, LFR and CFS modules.
> We will enlarge the text fonts and optimize the layout in both figures to ensure all details and tensor dimensions are readable.
>
> **Key Questions For Authors: The initialization of $\alpha$ in Eq.4 is not clearly specified**
> To ensure stable early-stage training, we apply a 'zero-initialization' strategy for the residual connection in Eq. 4. Specifically, we set the initial scalar $\alpha_{logit}$ to -1000, which yields a post-sigmoid gating weight $\alpha \approx 0$. Consequently, the initially refined prompt $P^G$ is numerically nearly identical to the base prompt $P$. This ensures that the introduction of the GIF module does not disrupt the initial training stability or corrupt the well-learned representation space of the frozen CLIP backbone. As training progresses, the network safely and gradually learns to incorporate the hierarchical context $Y$. Empirically, we observe that $\alpha$ converges to approximately 0.2 at the end of training. This demonstrates that the network effectively retains the base prompt $P$ as the dominant guidance signal, while utilizing the newly learned contextual prompt $Y$ as a fine-grained, approximately 20% structural rectification to robustly compensate for missing modalities.
>
> **Limitations: Lack of discussion regarding the proposed method's limitations**
> The reliance on a frozen backbone is a shared characteristic of the Parameter-Efficient Fine-Tuning and prompt learning paradigms. In future work, we plan to explore hybrid architectures that combine SPR with localized backbone unfreezing techniques, to find an optimal balance between leveraging the priors of the frozen network and enhancing model plasticity.

---

> > ### Author Rebuttal · Reviewer_17K1 · 2026-04-03
> >
> > Thank the authors for their response. I still have a few questions:
> >
> > * Regarding W2, the reported Hateful Memes (AUROC) results in Table 2 seem inconsistent with those in the original Table 2. Could the authors clarify this discrepancy?
> > * The discussion on limitations appears somewhat general. Could the authors provide some specific failure cases and analyze the possible reasons behind them?

---

> > > ### Author Response · Authors · 2026-04-04
> > >
> > > We are grateful for your professional questions. We address each point below.
> > >
> > > **Q1: Inconsistency of experimental data**
> > > We sincerely appreciate the reviewer’s careful scrutiny. This discrepancy arises from inconsistent random seed settings. Specifically, all core comparison experiments in Table 1 and W2 were conducted with the random seed set to 0. However, when conducting the experiments on the Hateful Memes dataset for Table 2, we unintentionally loaded a different configuration file with the random seed set to 1. We deeply apologize for this confusion. To verify the above explanation and further demonstrate the robustness of our proposed SPR network, we additionally conducted experiments on the Hateful Memes dataset under the 70% both-missing setting with five different random seeds. The corresponding experimental results are presented below.
> > > | Random Seed | &nbsp;&nbsp;&nbsp;&nbsp;&nbsp;&nbsp;SyP | &nbsp;&nbsp;&nbsp;&nbsp;&nbsp;Ours | Improvement |
> > > | :---: | :---: | :---: | :---: |
> > > | 0 | 68.42 | **71.08** | 2.66 |
> > > | 1 | 67.01 | **69.18** | 2.17 |
> > > | 2 | 70.31 | 73.58 | 3.27 |
> > > | 3 | 65.42 | 68.25 | 2.83 |
> > > | 4 | 71.14 | 74.19 | 3.05 |
> > > | Mean±Std | 68.46±2.34 | 71.26±2.62 | 2.8±0.42 |
> > >
> > > The results validate that our method achieves stable and consistent performance gains across all random seed settings. The full experimental logs are available via the anonymous link: https://anonymous.4open.science/r/Anonymous-data-87A6/. Furthermore, in the revised manuscript, we will update all experimental tables to report the mean and standard deviation across five random seeds, ensuring the rigor and transparency of our evaluation.
> > >
> > > **Q2: Detailed discussion of limitations**
> > > We thank the reviewer for the further inquiry. To investigate this, we manually analyzed 100 randomly selected error samples from the MM-IMDb dataset evaluated under the 90% Both-Missing scenario. We identified a recurring failure case when classifying *Horror* or *Thriller* movies under the Image-Missing setting. For instance, in sample '0057076', the plot is summarized as follows: "James willingly walks into a trap set with an innocent Russian beauty, solely to retrieve a Soviet encryption device stolen by SPECTRE". Textually, this shares immense semantic overlap with *Drama* or *Romance*. The distinguishing *Thriller* elements are typically conveyed via the dark, violent visual cues of the poster. When the poster is absent, SPR frequently misclassifies it as *Drama*.
> > > This phenomenon highlights the inherent limitations of prompting a frozen backbone under severe modality imbalance. Without visual corroboration, the frozen CLIP text encoder cannot dynamically reweight generic words (like *beauty* or *trap*) into genre-specific tropes. Furthermore, while our LFR module effectively ensures semantic consistency by grouping independent tokens into coherent units, it does not perform well in this specific case. By enforcing strict local coherence on phrases like "an innocent Russian beauty", LFR solidifies the text's benign, dramatic semantic structure, effectively over-smoothing subtle cues and preventing the sharp semantic shifts needed to activate the *Thriller* classification head. Finally, when the text lacks a genre-specific lexicon and the image is completely missing, the modality-specific linear projection $\mathrm{ProjDown}_m(P')$ in the GIF module processes a flat, non-discriminative vector. While the BiGRU captures sequential dependencies efficiently, it cannot map a generic text embedding to a visually-dominant label if the latent features are fundamentally absent. In future work, we plan to explore integrating external knowledge retrieval mechanisms to explicitly inject domain-specific priors. We will include this detailed discussion of limitations in the revised manuscript.

---

### Decision · Program_Chairs · 2026-04-30

**Decision:**

Accept (regular)

**Comment:**

All reviewers gave positive comments, so I recommend acceptance.